# Entropy Based Student’s *t*-Process Dynamical Model

**DOI:** 10.3390/e23050560

**Published:** 2021-04-30

**Authors:** Ayumu Nono, Yusuke Uchiyama, Kei Nakagawa

**Affiliations:** 1Graduated School of Engineering, The University of Tokyo, 7-3-1 Hongo, Bunkyo-ku, Tokyo 113-8656, Japan; 2MAZIN Inc., 3-29-14 Nishi-Asakusa, Tito City, Tokyo 111-0035, Japan; uchiyama@mazin.tech; 3NOMURA Asset Management Co. Ltd., 2-2-1 Toyosu, Koto-ku, Tokyo 135-0061, Japan; k-nakagawa@nomura-am.co.jp

**Keywords:** finance, volatility fluctuation, Student’s *t*-process, entropy based particle filter, relative entropy

## Abstract

Volatility, which represents the magnitude of fluctuating asset prices or returns, is used in the problems of finance to design optimal asset allocations and to calculate the price of derivatives. Since volatility is unobservable, it is identified and estimated by latent variable models known as volatility fluctuation models. Almost all conventional volatility fluctuation models are linear time-series models and thus are difficult to capture nonlinear and/or non-Gaussian properties of volatility dynamics. In this study, we propose an entropy based Student’s *t*-process Dynamical model (ETPDM) as a volatility fluctuation model combined with both nonlinear dynamics and non-Gaussian noise. The ETPDM estimates its latent variables and intrinsic parameters by a robust particle filtering based on a generalized H-theorem for a relative entropy. To test the performance of the ETPDM, we implement numerical experiments for financial time-series and confirm the robustness for a small number of particles by comparing with the conventional particle filtering.

## 1. Introduction

Asset allocation and pricing derivatives have been studied in both academia and industry as significant problems in financial engineering and quantitative finance. For these problems, various methodologies have been developed based on the variation of asset returns. In an idealized situation, the variation of returns has been assumed to follow the Gaussian distribution [1]. However, it is known that the variation of returns follows non-Gaussian distributions with fat tails [2]. To explain this observed fact, volatility, which quantifies the magnitude of fluctuating returns, has been introduced and utilized. Volatility, in particular, is often used as an indicator for constructing asset allocations that focus on macroeconomic fundamentals, and there are many studies related to them. Both researchers and investors have begun to attend to develop mathematical models of volatility fluctuations. For example, Yuhuang et al. investigated the impact of fundamental data on oil price volatility by focusing on time-varying skewness and kurtosis [3]. Hou et al. studied volatility spillovers between the Chinese fuel oil futures market and the stock index futures market, taking into account the time-varying characteristics of the markets [4].

In general, volatility is defined as the variance of a conditional Gaussian distribution for the variation of returns, namely, given as a latent variable in the literature of Bayesian statistical modeling. Based on this idea, various time-series models for the dynamics of asset returns have been developed and proposed. Such time-series models are generally called volatility fluctuation models, on which forecasting, state estimation and smoothing can be implemented.

In recent years, volatility fluctuation models with a machine learning technique have been proposed [5]. Since volatility is a latent variable, it is necessary for machine learning models to incorporate latent variables into their own methodology. The Gaussian process is a candidate, such as a Bayesian learning model [6], and its applications for several problems in finance have been reported [7,8]. The Student’s *t*-process is an extension of the Gaussian, for non-Gaussian distributed data such as asset returns. It has been proposed [9] and applied to the analysis of financial time-series and asset allocations, and it is confirmed for this model to estimate robustly [10].

This study extends the Student’s *t*-process latent variable model to a dynamic latent variable model incorporating the structure of time-series. To estimate dynamic latent variables, we used the particle filter method [11]. The particle filter is used to estimate the latent variables. Conventional particle filters have problems called weight bias and the particle impoverishment problem (PIP), directly affecting the estimation accuracy [12]. Then, the merging particle method [13] and Monte Carlo filter particle filter [14] have been proposed. However, these methods are computationally expensive because they need a large number of particles. Therefore, we used an Entropy-based particle filter (EBPF), which constructs a parametric prior distribution on the generalized H-theorem for relative entropy [15]. It is expected to prevent the bias of particle weights and the loss of particle diversity while reducing the computational cost. Using EBPF in this experiment, and comparing it with conventional methods, we confirmed that it is effective for finance problems.

In summary, to estimate robustly and avoid the particle filter’s problem, we combined *t*-process dynamical model and EBPF. We call the proposed model an entropy based Student’s *t*-process dynamical model (ESTDM), in the following. We will verify this model’s usefulness. The remains of this paper are summarized as follows—Section 2 introduces related statistical and machine learning models. In Section 3, we derive and propose ESTDM with its filtering method. In Section 4, we show the performance of volatility estimation using the proposed method and discuss the results. Section 5 is devoted to our conclusions and future perspectives.

## 2. Related Work

### 2.1. Volatility Fluctuation Models

One of the most basic and utilized volatility fluctuation models is the GARCH model [16] given as follows: (1)$$\begin{array}{c} x_{t}∼N\left(0,\sigma_{t}^{2}\right), \end{array}$$(2)$$\begin{array}{c} \sigma_{t}^{2}=\alpha_{0}+\sum_{j=1}^{q}\alpha_{j}\sigma_{t-j}^{2}+\sum_{i=1}^{p}\beta_{i}x_{t-i}^{2}, \end{array}$$
where $x_{t}$ is a time-dependent random variable sampled from a Gaussian distribution with mean 0 and variance $\sigma_{t}^{2}$, and the time evolution of the variance is given by Equation (2). The parameters $\alpha_{j}$ and $\beta_{i}$ take positive values, which can be estimated by observed data. Positive integers *p* and *q* are the order of the regression, respectively. Then this model is known as the GARCH(*p*, *q*) model. For the sake of simplicity, the order parameters are often fixed as p=q=1. Various families of GARCH model have been developed and proposed in the area of econometrics and quantitative finance [17]. For instance, asymmetric effect has been introduced into a multivariate GARCH model [18,19].

### 2.2. Gaussian Process

For any finite number of vectors $\left\{x_{1},x_{2},\cdots ,x_{n}\right\}$ and a stochastic process f(·), if the joint probability density function $\left\{f\left(x_{1}\right),f\left(x_{2}\right),\cdots ,f\left(x_{n}\right)\right\}$ is a Gaussian distribution, f(·) is called a Gaussian process [6]. Since the Gaussian process samples an infinite-dimensional vector, the mean value function m(·) and the covariance function K(·,·) are introduced as follows: (3)$$\begin{array}{c} m(x)=E[f(x)], \end{array}$$(4)$$\begin{array}{c} K\left(x,x^{\prime }\right)=E\left[\left(f\left(x\right)-m\left(x\right)\right){\left(f\left(x^{\prime }\right)-m\left(x^{\prime }\right)\right)}^{T}\right]. \end{array}$$
Then, given a matrix $X={\left[x_{1},x_{2},\cdots ,x_{n}\right]}^{T}$, p(f|X)=N(m(X),K(X,X)) is the probability density function of the Gaussian process. When we explicitly state that the stochastic process *f* is sampled from the Gaussian process, we write f∼GP(m,K). Without loss of generality, the mean function of the Gaussian process is often assumed to be zero. The covariance function is represented by the kernel function k(·,·), which is a positive symmetric bi-variate function, satisfying
(5)$$K\left(x,x^{\prime }\right)=k\left(x,x^{\prime }\right).$$
Hence, K(X,X) is a positive definite symmetric matrix. As a kernel function, for example, the radial basis function
(6)$$k_{\mathrm{RBF}}\left(x,x^{\prime }\right)=\alpha \exp(-l^{-2}||x-x^{\prime }{\left|\right|}^{2})$$
is often used. Here, α and *l* are hyper parameters.

For a pair of observed data $D=\{\left(x_{1},y_{1}\right),\left(x_{2},y_{2}\right),\cdots ,\left(x_{n},y_{n}\right)\}$, let $X={\left[x_{1},x_{2},\cdots ,x_{n}\right]}^{T}$, $Y={\left[y_{1},y_{2},\cdots ,y_{n}\right]}^{T}$. The hyper parameters of the Gaussian process can be estimated by gradient and Monte Carlo methods on D. From the trained Gaussian process, the prediction $Y^{*}={\left[y_{1}^{*},y_{2}^{*},\cdots ,y_{m}^{*}\right]}^{T}$ for unknown input $X^{*}={\left[x_{1}^{*},x_{2}^{*},\cdots ,x_{m}^{*}\right]}^{T}$ is sampled from the conditional Gaussian distribution $N(f^{*},K^{*})$. The mean function $f^{*}$ and the covariance function $K^{*}$ of the conditional Gaussian distribution are given by
(7)$$\begin{array}{c} f^{*}=m_{X}+K_{X^{*},X}K_{X,X}^{-1}Y, \end{array}$$
(8)$$\begin{array}{c} K^{*}=K_{X^{*},X^{*}}-K_{X^{*},X}K_{X,X}^{-1}K_{X,X^{*}}. \end{array}$$
It is seen that the mean and covariance functions of the Gaussian process propagate the information of previously observed data to predicted values.

### 2.3. Student’s t-Process

In the Gaussian process, it is assumed for the probability density function to be the Gaussian distribution. Thus, when we apply the Gaussian process to data following a probability distribution with fat tails, such as financial time-series, it is impossible to perform an accurate estimation. A model that extends the Gaussian process to such data is the Student’s *t*-process [9]. The Student’s *t*-process is a stochastic process f(·) with ν degrees of freedom and a Student’s *t*-distribution defined as follows:(9)$$T\left(m,K,\nu \right)=\frac{\Gamma \left(\frac{\nu +n}{2}\right)}{{\left[\left(\nu -2\right)\pi \right]}^{\frac{n}{2}}\Gamma \left(\frac{\nu }{2}\right){\left|K_{X,X}\right|}^{\frac{1}{2}}}{\left[1+\frac{1}{\nu -2}{\left(Y-m_{X}\right)}^{T}K_{X,X}^{-1}\left(Y-m_{X}\right)\right]}^{-\frac{\nu +n}{2}}.$$
Here, m(·) and K(·,·) are the mean and covariance functions, respectively, and Γ(·) is the gamma function. When the stochastic process f(·) is a Student’s *t*-process, it is denoted by f∼TP(m,K;ν). As with the Gaussian process, the mean function of the Student’s *t*-process is often assumed to be zero without loss of generality.

The predictive distribution of the Student’s *t*-process is also the Student’s *t*-distribution $T(m^{*},K^{*},\nu^{*})$, where degrees of freedom, mean and covariance functions are then updated as follows: (10)$$\begin{array}{c} \nu^{*}=\nu +n, \end{array}$$(11)$$\begin{array}{c} m^{*}=m_{X}+K_{X^{*},X}K_{X,X}^{-1}\left(Y-m_{X}\right)\\ K^{*}=\frac{\nu -\beta -2}{\nu -n-2}\left[K_{X^{*},X^{*}}-K_{X^{*},X}K_{X,X}^{-1}K_{X,X^{*}}\right] \end{array}$$(12)$$\begin{array}{c} \beta ={\left(Y-m_{X}\right)}^{T}K_{X,X}^{-1}\left(Y-m_{X}\right). \end{array}$$
Unlike the Gaussian process, in the Student’s *t*-process, we can confirm that the effect of the number of data is reflected in the update equations of the degrees of freedom and the covariance function.

### 2.4. Student’s t-Process Latent Variable Model

In the Student’s *t*-process latent variable model, the input matrix *X* is given as a latent variable. Assume that the observed data $y\in R^{D}$ and the latent variable $x\in R^{Q}$ are related as y=f(x) by the Student’s *t*-process f∼TP(m,K;ν). When we let $Y\in R^{D\times N}$ be the sequence of *N* observed data, and $X\in R^{Q\times N}$ be the sequence of *N* latent variables, we can define the following model as Student’s *t*-process latent variable model [10]:(13)$$p\left(Y|X\right)=\frac{\Gamma \left(\frac{\nu +D}{2}\right)}{{\left[\left(\nu -2\right)\pi \right]}^{\frac{D}{2}}\Gamma \left(\frac{\nu }{2}\right){\left|K_{X,X}\right|}^{\frac{1}{2}}}{\left[1+\frac{1}{\nu -2}{\left(Y-m_{X}\right)}^{T}K_{X,X}^{-1}\left(Y-m_{X}\right)\right]}^{-\frac{\nu +D}{2}}.$$
Since the Student’s *t*-distribution converges to the Gaussian distribution in the limit of ν→∞, we can see that the Student’s *t*-process latent variable model embraces the Gaussian process latent variable model [20].

## 3. Proposed Model

### 3.1. Student’s *t*-Process Dynamical Model

Since volatility fluctuations cannot be observed directly, they are modeled as dynamic latent variables, such as the family of GARCH models, most of which are given by linear time-series models [18]. To describe nonlinear dynamics of volatility fluctuations, we extend the Student’s *t*-process latent variable model to dynamic latent variables, namely, Student’s *t*-process dynamical model (TPDM), which is expected to be robust for both observable and unobservable with outliers.

Suppose $p_{t}$ represents an asset price at time *t*, the log-return is given by $r_{t}=\log\left(p_{t}/p_{t-1}\right)$. Let $\sigma_{t}^{2}$ denote the volatility of $r_{t}$. Here, for an observable $r_{t}$ and a latent variable $\sigma_{t}^{2}$, we provide a volatility fluctuation model by a TPDM as follows: (14)$$\begin{array}{c} r_{t}∼T\left(0,\sigma_{t}^{2};\nu \right), \end{array}$$(15)$$\begin{array}{c} v_{t}\equiv \log\sigma_{t}^{2}, \end{array}$$(16)$$\begin{array}{c} v_{t}=f\left(v_{t-1},r_{t-1};\nu \right)+\epsilon_{t} \end{array}$$(17)$$\begin{array}{c} \epsilon_{t}∼N\left(0,\sigma_{n}^{2}\right), \end{array}$$
where the observable $r_{t}$ as centered at 0 and following a Student’s *t*-distribution with ν degrees of freedom, whose parameter is given by $\sigma_{t}^{2}$. The dynamic latent variable $v_{t}$ is defined by Equation (15) to take the whole real number as its range of value. The time evolution of the dynamic latent variable $v_{t}$ is given by Equation (16) with a Gaussian white noise whose variance is $\sigma_{n}$. The stochastic process *f* on the right-hand side of Equation (16) follows a Student’s *t*-process given by
(18)$$\begin{array}{c} f∼\mathrm{TP}(m,K;\nu ), \end{array}$$
(19)$$\begin{array}{c} m\left(\xi_{t-1}\right)=av_{t-1}+bx_{t-1}, \end{array}$$
(20)$$\begin{array}{c} k\left(\xi_{t-1},\xi_{t-1}^{\prime }\right)=\gamma \exp\left(-l^{-2}||\xi_{t-1}-\xi_{t-1}^{\prime }|{|}^{2}\right), \end{array}$$
where $\xi_{t}=\left(x_{t},v_{t}\right)$, and the hyper parameters are $\theta =(\nu ,\sigma_{n},a,b,\gamma ,l)$. Given a series of observed data $r_{1:T}=\left\{r_{1},r_{2},\cdots ,r_{T}\right\}$, it is possible to obtain the volatility fluctuations by estimating a series of dynamic latent variables $v_{1:T}=\left\{v_{1},v_{2},\cdots ,v_{T}\right\}$.

### 3.2. Particle Filter

Particle filter is a method of state estimation by Monte Carlo sampling, where a large number of particles approximates posterior distributions. Hence, it can be applied to nonlinear systems, where posterior distributions are intractable [21]. For *N* particles, let ${\left\{v_{1:t-1}^{i}\right\}}_{i=1}^{N}$ and $W_{t-1}^{i}\;\left(i=1,2,\cdots ,N\right)$ be the realizations of the dynamic latent variables and their associated weights up to time t−1, respectively. The weights are normalized to $\sum_{i=1}^{N}W_{t-1}^{i}=1$. With these values, the posterior distribution $p(v_{1:t-1}|x_{1:t-1})$ at time t−1 can be approximated as follows [22]:(21)$$\hat{p}\left(v_{1:t-1}|x_{1:t-1}\right)=\sum_{i=1}^{N}W_{t-1}^{i}\delta \left(v_{1:t-1}\right),$$
where δ(·) is the Dirac’s delta function. In other words, the posterior distribution is approximated by a mixture of the delta functions.

It is however known that an insufficient number of particles fails to approximate the posterior distribution by the degeneracy of ensemble. Indeed, each particle’s weights become unbalanced and biased toward a tiny number of particles as the time step progresses [11,12]. To overcome this problem, a huge amount of particles is used for filtering processes in practice.

### 3.3. Entropy-Based Particle Filter

In the use of the conventional particle filter, it is necessary to sample a large number of particles for accuracy. That leads to the growth of estimation time. In the case of online estimation, reducing run time is desired. For this purpose, we introduce a robust particle filter for a small number of particles.

Let us reconsider approximating the probability density function for the dynamic latent variable, Q(v,t), called a background distribution. In the conventional particle filter, the background distribution is approximated by the mixture of delta functions. This approximation works well only when the background distribution exhibits an extensively sharp peak. Nevertheless, the delta function has no width, and the distribution peaks only at a single point.

To improve the accuracy for the approximation of the background distribution, we replace the mixture of the delta functions with that of Gaussian distributions as
(22)$$\hat{Q}\left(v,t\right)=\sum_{i=1}^{M}W_{t}^{i}N\left(\mu_{t}^{i},\sigma_{t}^{2,i}\right),$$
where $\mu_{t}^{i},\sigma_{t}^{2,i}\;\left(1\leq i\leq M\right)$ are the mean and variance of the Gaussian distributions at *t*. Unlike the delta function, the Gaussian distribution has a certain width in its distribution. Hence, the mixture of the Gaussian distributions is capable of fitting properly to data with large variance and fat tails.

With the use of finite samples from the background distribution Q(v,t), the posterior/filter distribution P(v,t) is inferred by the minimum principal for relative entropy, which is known as an entropy based particle filter [15]. The relative entropy (Kullback-Liebler divergence) between the filter distribution P(v,t) and the background distribution Q(v,t) are defined as follows [23,24,25]:(23)$$H\left[P|Q\right]=\int_{\Omega_{v}}P\left(v,t\right)\log\left(\frac{P(v,t)}{Q(v,t)}\right)dv,$$
where $\Omega_{v}$ is the domain of the dynamic latent variable $v_{t}$. On the properties of the relative entropy as a quasi-distance for probability density functions, the filter distribution is obtained as the minimizer for the relative entropy in Equation (Equation 23). Combined with the entropy based particle filter, the state estimation of the ESTDM is implemented. An overview of its algorithm is explained in the following Algorithm 1.
**Algorithm 1** Entropy Based Student’s *t*-Process Dynamical Model (ETPDM)**Require:** Initial particles $X_{0}=\left[X_{0}^{0},...,X_{0}^{M}\right]$, Initial particles’ weights $W_{0}^{i}=1/M$**Ensure:** $\sum_{i=1}^{M}W_{t}^{i}=1.0$ at any time *t*1:**while** There are observations to be assimilated **do**2: Compute importance weights proportional to the likelihood with observation $x_{t}$
(24)$$W_{t}^{i}\propto p\left(y_{t}|X_{t}\right)$$ According to weights $\left\{W_{t}^{i}\right\}$, resample *M* particles ${\left\{X_{t}^{j}\right\}}_{j=1}^{M}$. Then we can compute filter distribution $Q^{\prime }\left(x\right)$ at time *t*
(25)$$Q_{t}^{\prime }=\sum_{i=1}^{M}W_{t}^{j}N\left(X_{t}^{j}\right).$$ At the same time, we’re also able to estimate expected status $v_{t}$ , extracting any finite number of samples $\left\{v^{k}\right\}$ from background density $Q_{t}$
(26)$$v_{t}=E\left(v^{k}\right).$$ With stochastic process f∼T(m,k;ν), generate new particles
(27)$$\left\{X_{t+1}^{i}\right\}=f\left(X_{t}^{j},x_{t}\right)$$ Then we can predict distribution $\hat{Q}\left(x\right)$ at time t+1
(28)$${\hat{Q}}_{t+1}=\sum_{i=1}^{M}W_{t}^{i}N\left(X_{t+1}^{i}\right).$$3: **return** Log likelihood for estimation $p(y_{t}|v_{t})$.4:**end while**

## 4. Numerical Experiments

In this section, we implement numerical experiments to validate the ETPDM for the time-series of a foreign exchange rate. As a dataset, we use USD/JPY exchange rate in 2010—every 1-min sampled, 30-min sampled and 1-h sampled. Figure 1, Figure 2 and Figure 3 show the time-series of the log-return of the USD/JPY exchange rate $r_{t}$ and volatility fluctuations estimated by respective the ETPDM, the conventional particle filter for the GARCH model (cp-GARCH) and the conventional particle filter for the Student’s *t*-process dynamical model (cp-TPDM). Warm up period of the estimations is 0≤t≤20, where the values of volatility show zeros. In Figure 1a, intermittent jumps are observed, which are evidence of the non-Gaussian behavior of $r_{t}$. Indeed, the estimated volatility fluctuations show higher peaks at the same time point of the intermittent jumps in Figure 1b–d. That means all of the models capture the nature of volatility fluctuations of the USD/JPY exchange rate effectively. Besides, the same can be said for other types of data sets—30-min and 1-h—in Figure 2 or Figure 3, which means that these models can be applied to data of any sampling rate. Previous volatility estimation studies used the GARCH model with various estimation methods. A typical example is the particle filter [26], or the Markov chain Monte Carlo simulation [27]. In all of these studies, including this experiment, the GARCH model has been implemented well.

Figure 4 show the estimated log-likelihoods of the ETPDM, cp-GARCH and the cp-TPDM. The likelihood tends to be higher for the ETPDM, the cp-TPDM and cp-GARCH in that order. As mentioned in Section 2, TPDM is a superordinate model that encompasses Gaussian process dynamical model, the GPDM, and the fact that the likelihood of the GARCH was at a lower level than that of TPDM is consistent with the results of previous studies comparing the GARCH and the GPDM [26]. Likelihood is the most reliable indicator to quantify the model performance, and the EPTDM had the best performance of three models. Besides, in the case of cp-TPDM, the log-likelihoods scatter around −0.7 in the range of particle numbers from 10 to 500 without convergence. This means the performance of the conventional particle filter is insufficient for the given data. On the other hand, the log-likelihoods of the ETPDM exhibit good convergence for the particle numbers larger than 100 in Figure 4b, which indicates the ETPDM is expected to be robust for fewer sampling.

To investigate the effectiveness of particle filtering, we introduce an effective particles rate
(29)$$R_{eff}=\frac{1}{N\sum_{i=1}^{N}{\left(W_{t}^{i}\right)}^{2}}$$
as a measure for evaluating the bias of sampled particles. This value gives the maximum value $R_{eff}=1$ when the weights are uniformly distributed as $W^{i}=1/N\;\left(i=1,2,\ldots ,N\right)$. In Figure 5a, the effective particle rates for the cp-TPDM scatter for whole particle numbers from 10 to 500. This kind of worse performance for the effective particle rates stems from the weight bias problem of the particle filtering. In other words, the conventional particle filter is hard to overcome the particle impoverishment problem, even if, by increasing particle numbers. On the contrary, for the case of the ETPDM as shown in Figure 5b, we can see that the effective particle rate converges beyond 50%. This is the expected advantage of the ETPDM, which stems from the finite band of each Gaussian distribution as a component of the prior distribution. Thus, the ETPDM serves as an accurate estimation for lower particle numbers and then would contribute to effective online estimation. Focusing on another comparison of the ETPDM, the cp-GARCH, also looks good in the view of the effective particles rate. However, when we also consider the likelihood in Figure 4a, we can say that the practical ensembles didn’t affect to performance because cp-GARCH was less suitable for this problem than the ETPDM. This is another evidence that the ETPDM have better potential.

In order to validate the robustness for estimating intermittent return dynamics, we investigate the degree of freedom ν of the Student’s *t*-process dynamical models. For this purpose, we split time window of return fluctuations; one is low volatility window (50≤t≤250) and the other is high volatility counterpart (360≤t≤560). The descriptive statistics of the return fluctuations in the two windows are shown in Table 1. As is seen in the table, kurtoses in both time windows are larger than 3, namely, corresponding return fluctuations follow non-Gaussian statistics. Prior research has confirmed that when the data set follows a Gaussian distribution, the strengths of models that excel at robust estimation do not come into play [28]. Therefore, such a data set that follows a non-Gaussian distribution is appropriate for the purpose of this experiment. Figure 6 exhibit the log-likelihoods of the cp-TPDM and the ETPDM in (a) low volatility window and (b) high volatility one. In these figures, the log-likelihoods of the ETPDM in both time windows have maxima in 6≤ν≤7 though the estimations by the the cp-TPDM are unstable. This result evidences the robustness of the ETPDM.

## 5. Conclusions

In this study, we proposed the ETPDM to implement robust estimation for dynamical latent variables of nonlinear and non-Gaussian fluctuations. In estimating the dynamic latent variables and hyper parameters, the entropy based particle filter with the Gaussian mixture distribution was adopted. To validate the performance of the ETPDM, we carried out the numerical experiment for the return fluctuations of a foreign exchange rate compared with the cp-GARCH and the cp-TPDM. As a result, we confirmed the advantages of the ETPDM; (i) good convergence property, (ii) high effective particle rate and (iii) robustness for a small number of particles.

Based on its advantages, the ETPDM is applicable for online volatility estimation for the problem of asset allocation and derivative pricing in a short time span. As a basis distribution for background distribution, we employed the Gaussian distribution in our numerical experiments. Nevertheless, the framework of the entropy based particle filter is able to be extended to other probability density functions. Additionally, we can adapt this research to any other time-series data, not just asset data. It has the potential to be applied to control engineering, such as the self-positioning estimation problem. These are our future works.

## Figures and Tables

**Figure 1 entropy-23-00560-f001:**
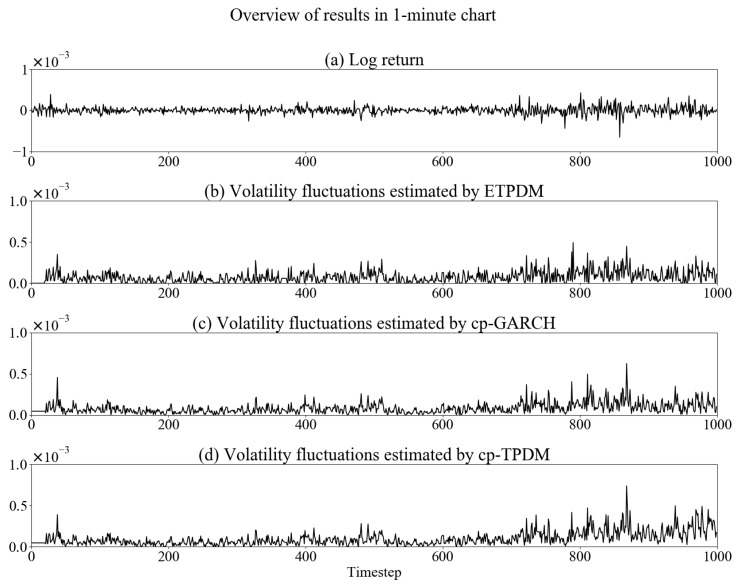
Overview of estimation results in 1-min chart.

**Figure 2 entropy-23-00560-f002:**
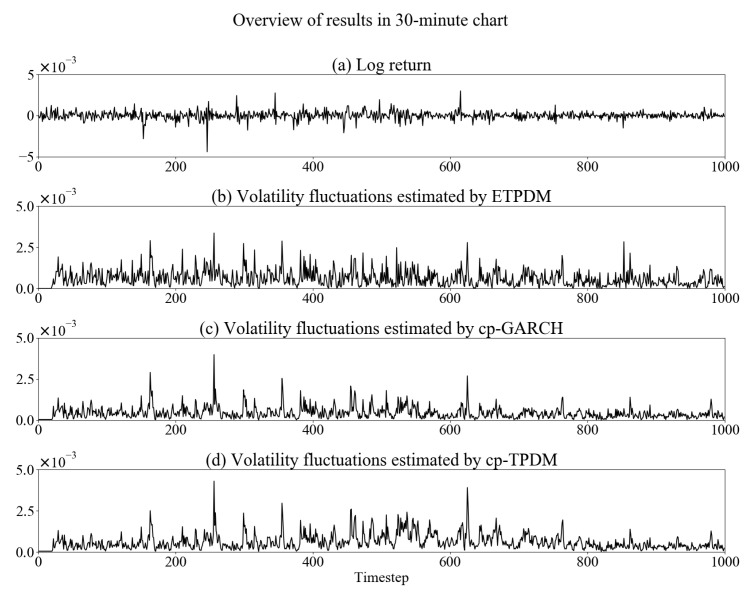
Overview of estimation results in 30-min chart.

**Figure 3 entropy-23-00560-f003:**
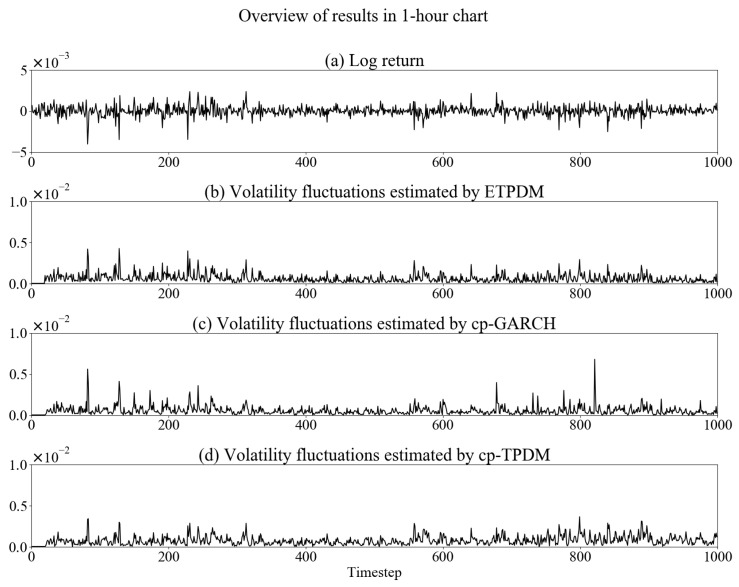
Overview of estimation results in 1-h chart.

**Figure 4 entropy-23-00560-f004:**
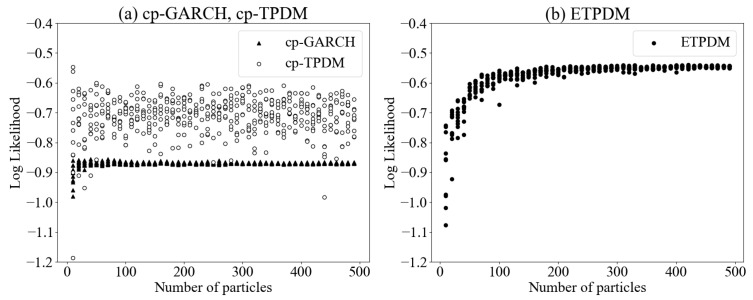
Estimated log-likelihoods of (**a**) the cp-GARCH, the cp-TPDM and (**b**) the ETPDM.

**Figure 5 entropy-23-00560-f005:**
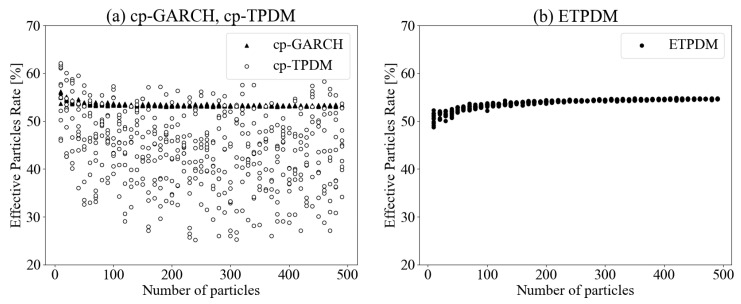
Effective particle rates of (**a**) cp-GARCH, the cp-TPDM and (**b**) the ETPDM.

**Figure 6 entropy-23-00560-f006:**
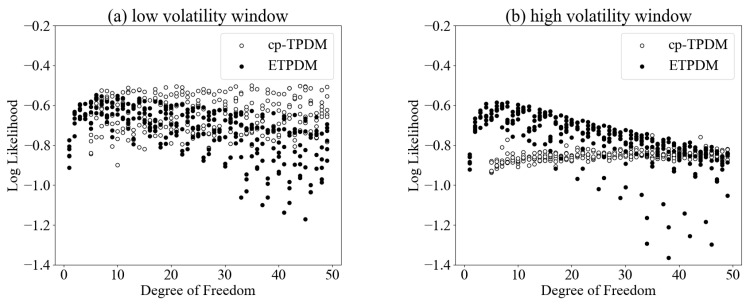
Log-likelihoods of the cp-TPDM and the ETPDM in (**a**) low volatility window and (**b**) high volatility one.

**Table 1 entropy-23-00560-t001:** Two types of window.

Window Type	Mean	Variance	Skewness	Kurtosis
high volatility window	$-1.0\times 10^{-6}$	$8.31\times 10^{-4}$	−0.974	5.22
low volatility window	$1.0\times 10^{-6}$	$4.41\times 10^{-4}$	−0.531	3.56

## Data Availability

The data that support the findings of this study are available from the corresponding author, Nono, A., upon reasonable request.

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
