# Peer review of "Entropy Based Student’s t-Process Dynamical Model"

_entropy, 2021, doi:10.3390/e23050560_

Round 1

Reviewer 1 Report

My comments are included in the attached  .pdf document.

Reviewer 2 Report

The paper proposes  an entropy based Student’s t-process to implement robust estimation for dynamical latent variables of nonlinear and non-Gaussian fluctuations.

It is a very interesting and well-presented paper.

1).  It would be interesting if authors would show an analytic expression for the expected improvement under a Student’s-T distribution from eq. (9).

2). I would suggest the authors to include relative reference from Entropy if possible.

3). In the Introduction section, 3rd paragraph, other works could be mentioned as well. 

Round 2

Reviewer 1 Report

no additional comments